# The transcription factor FOXL2 mobilizes estrogen signaling to maintain the identity of ovarian granulosa cells

Adrien Georges[1,2‡], David L'Hôte[1,2†§], Anne Laure Todeschini[1,2*†], Aurélie Auguste[1,2], Bérangère Legois[1,2], Alain Zider[1,2], Reiner A Veitia[1,2*]

[1]Institut Jacques Monod, Paris, France; [2]Université Paris Diderot, Paris VII, Paris, France

**Abstract** FOXL2 is a lineage determining transcription factor in the ovary, but its direct targets and modes of action are not fully characterized. In this study, we explore the targets of FOXL2 and five nuclear receptors in murine primary follicular cells. We found that FOXL2 is required for normal gene regulation by steroid receptors, and we show that estrogen receptor beta (ESR2) is the main vector of estradiol signaling in these cells. Moreover, we found that FOXL2 directly modulates *Esr2* expression through a newly identified intronic element. Interestingly, we found that FOXL2 repressed the testis-determining gene *Sox9* both independently of estrogen signaling and through the activation of ESR2 expression. Altogether, we show that FOXL2 mobilizes estrogen signaling to establish a coherent feed-forward loop repressing *Sox9*. This sheds a new light on the role of FOXL2 in ovarian maintenance and function.

**\*For correspondence:**
todeschini@ijm.univ-paris-diderot.fr (ALT); veitia.reiner@ijm.univ-paris-diderot.fr (RAV)

†These authors contributed equally to this work

**Present address:** ‡Equipe Régulation transcriptionnelle des génomes, CEA Saclay, Gif-sur-Yvette, France; §Unité de Biologie Fonctionnelle et Adaptative, UMR, Paris, France

## Introduction

FOXL2 is a key transcriptional regulator of the differentiation and maintenance of granulosa cells, those supporting oocyte maturation and growth during folliculogenesis (*Schmidt et al., 2004*; *Uhlenhaut et al., 2009*; *Georges et al., 2014*). Indeed, mature granulosa cells are absent in *Foxl2⁻/⁻* mice because pre-granulosa cells remain blocked and do not undergo further differentiation (*Schmidt et al., 2004*). Moreover, depletion of *Foxl2* in already mature granulosa cells in mice results in their (molecular) trans-differentiation and in the upregulation of markers of Sertoli cells, the male counterpart of granulosa cells (*Uhlenhaut et al., 2009*). In humans, FOXL2 heterozygous mutations are responsible for the blepharophimosis-ptosis-epicanthus inversus syndrome (BPES), characterized by facial malformations often associated with primary ovarian insufficiency (POI) (*Crisponi et al., 2001*). Recent works have shown that FOXL2 mutations impairing its DNA binding and/or transcriptional activity are responsible for POI occurrence in BPES (*Dipietromaria et al., 2009*; *Todeschini et al., 2011*). Interestingly, a specific somatic mutation of FOXL2 has been identified in more than 95% of adult-type granulosa cell tumors (GCTs) confirming the strong association of FOXL2 with granulosa cell fate and function (*Shah et al., 2009*; *Jamieson and Fuller, 2012*).

Despite this accumulating wealth of knowledge, the mechanisms by which FOXL2 regulates granulosa cell differentiation are not well understood. This is in particular due to the difficulty to study the pre-granulosa-to-granulosa cell transition. FOXL2 transcriptional targets have been studied using various models such as GCT-derived cell lines or ovaries from constitutive knockout mice, preventing thus far to fully understand the function of FOXL2 in healthy granulosa cells (*Georges et al., 2014*). However, there are some well established facts such as the cooperation between FOXL2 and SMAD3 on several enhancers, in particular one driving the expression of follistatin, a key factor of ovarian function (*Blount et al., 2009*; *Tran et al., 2011*, 3). FOXL2 has also been described to interact with the

**eLife digest** In female mammals, granulosa cells in the ovaries help egg cells to grow and develop by secreting nutrients and estrogens—the female sex hormones. A protein called FOXL2 helps granulosa cells to develop and functions by binding to the DNA of the cells to switch certain genes either on or off.

In humans, mutations in the gene that codes for the FOXL2 protein are associated with granulosa cell tumors and with a loss of female fertility in early adulthood. In addition, if the amount of FOXL2 is artificially reduced in granulosa cells in female mice, the cells take on many of the characteristics of supporting cells found in the testes of males.

To investigate in more detail how FOXL2 works, Georges et al. grew mouse granulosa cells in the laboratory to identify the DNA sequences where FOXL2 will bind, and to uncover how this binding affects gene expression. Georges et al. conclude that FOXL2 orchestrates a network involving many different proteins that allows estrogen to be produced and used by granulosa cells; and in doing so these cells maintain their identity as ovarian cells. FOXL2 was also shown to work closely with the receptor proteins that detect the sex hormones, and which help to control whether particular sex-specific genes are switched on or off.

One particularly important role of FOXL2 in granulosa cells is that it represses a gene called *Sox9*. By repressing *Sox9*, the granulosa cells do not transform into their counterparts normally found in testes. Although FOXL2 was previously reported to directly regulate the *Sox9* gene, Georges et al. find that it also acts through other molecules, and that there are alternative ways in which it can do so.

Although Georges et al. have established some of the ways that FOXL2 functions, this protein can work via other pathways; these will require further investigation to be fully understood.

estrogen receptor alpha (ESR1) in mouse ovaries to repress the expression of *Sox9*, an essential transcriptional regulator of Sertoli cells (*Uhlenhaut et al., 2009*). Interestingly, mice depleted of both estrogen receptors (ESR1 and ESR2) display a transdifferentiation of granulosa cells apparently similar to that observed when *Foxl2* is depleted in mature cells (*Couse et al., 1999*). FOXL2 interacts with many other transcription factors of the nuclear receptor (NR) superfamily, such as NR5A1 (also known as Steroidogenic factor 1, SF-1), an orphan NR critical for gonadal development; NR2C1, another orphan NR whose paralog NR2C2 is involved in folliculogenesis, and the progesterone receptor PGR (*Park et al., 2010*; *L'hôte et al., 2012*; *Ghochani et al., 2012*). This led us to hypothesize that FOXL2 might be involved in coordinating NR activity in granulosa cells.

Here, we explore the transcriptional targets of FOXL2 and its impact on target gene regulation by five NRs (ESR1, ESR2, AR, NR5A1, and NR2C1). This work was performed in murine primary follicular cells using a knockdown approach coupled to high-throughput genomic technologies. These analyses allowed us to uncover a thorough core of FOXL2 transcriptional targets in preantral-small antral follicles, which includes many key genes of ovarian function. A ChIP-Seq analysis of FOXL2 genomic binding sites in these cells suggests that it mainly binds to regions located in the introns of its targets, and provides a resource of potential regulatory elements. We find that FOXL2 strongly influences gene regulation by estrogen and androgen receptors. Moreover, we show that FOXL2 is required for normal expression of *Esr2* through a newly identified *Esr2* intronic regulatory element and that ESR2 is the main effector of 17β-estradiol (E2) signaling in these cells. FOXL2 is also required for the expression of *Cyp19a1*, which encodes the limiting enzyme in estrogen production, and FOXL2, ESR2 and estradiol all participate in SOX9 repression. Our data show that FOXL2 expression establishes a coherent subnetwork allowing estrogen production and signaling in granulosa cells as well as the maintenance of granulosa identity through an indirect repression of SOX9.

## Results

### Transcriptomic effects of the depletion of FOXL2 and various NRs

To determine the transcriptional targets of FOXL2, we set up a model of murine primary granulosa cells for performing knockdown and transcriptomic analyses. Specifically, we purified preantral-small

antral follicles from the ovaries of 8-week-old female Swiss mice. After an overnight culture, the follicles were dispersed using trypsin and cells were grown for a maximum of 8 days. Follicular cells are known to progressively (trans)differentiate in culture, by upregulating markers of Sertoli cells or by luteinizing (*Nekola and Nalbandov, 1971*). We therefore verified by immunofluorescence that the expression of FOXL2, which is absent in luteinized cells, was maintained throughout culture and that SOX9 was not substantially upregulated. We observed a similar level of FOXL2 expression in follicles cultured for one night as in 9-day cultured cells (*Figure 1A*). SOX9 upregulation was observed in about 20% of the cells. However, we could check that the known FOXL2-mediated repression of *Sox9* was still operant (*Figure 1B*). Indeed, upregulation of SOX9 could be detected as soon as 24 hr after treating the cells with a commercial siRNA pool targeting FOXL2, and its expression continued to increase 48 hr after treatment. This stresses the importance of limiting the interval between knockdown and RNA analysis to assess the direct and indirect targets of FOXL2, as transcriptional changes can also occur because of SOX9 upregulation. We also verified that we could detect the expression of six NRs of interest (*Figure 1—figure supplement 1*). We observed that *Ar, Esr2* and *Nr5a1* were the most expressed NRs, and in particular that *Esr2* was at least four times more expressed than *Esr1* in these cells, in agreement with previous observations (*Lenie and Smitz, 2008*). Since progesterone receptor was very poorly expressed, we removed it from the analysis. We focused instead on the targets of the androgen receptor, which is required for terminal folliculogenesis (*Walters et al., 2012*).

Next, we treated the cultured primary follicular cells with commercial siRNA pools targeting *Foxl2* and/or *Esr1, Esr2, Ar, Nr5a1* or *Nr2c1*. We verified siRNA efficiency by qPCR (*Figure 1—figure supplement 2*). We observed a systematic decrease of more than 70% of the targeted mRNAs. We also noticed that *Esr2* expression was reduced by about 50% following *Foxl2* knockdown, suggesting that FOXL2 activates *Esr2* expression. As expected, *Sox9* expression was repressed by both FOXL2 and ESR2, as it increased in the absence of either of them, and even more in the absence of both. On the contrary, NR5A1 was required to activate *Sox9* expression. These results recapitulate well-known in vivo findings (*Kato et al., 2012*; *Uhlenhaut et al., 2009*) and confirm these primary cells are suitable to study the transcriptional regulation by FOXL2 and its interplay with NRs.

We next performed a microarray analysis of the transcriptional changes induced by FOXL2 or NR knockdowns. Specifically, we treated cells on the one hand with control or FOXL2-targeting siRNAs and on the other hand with control siRNAs or siRNAs targeting each of the NRs. This experimental design gave us six different conditions in which we could study the transcriptional targets of FOXL2 independently of the action of the NRs. We thus used a paired *Significance Analysis of Microarrays* (*Tusher et al., 2001*). This analysis detected 863 genes whose expression was modified by FOXL2 with a median false discovery rate <0.1%. Among them, we selected the genes undergoing an expression fold-change of at least 1.5. We thus obtained a list of 224 FOXL2-activated targets (i.e., with decreased expression in the absence of FOXL2) and 139 repressed genes (*Figure 1C* and *Figure 1—source data 1*). This list contained previously known targets of FOXL2 such as *Fst* (follistatin); *Ptgs2*, encoding a cyclo-oxygenase required for folliculogenesis; *Fos*, coding for a subunit of the AP-1 transcription factor; *Cyp19a1*, encoding the aromatase enzyme; and *Bmpr1b*, coding for a BMP receptor. Many other genes known to be critical for folliculogenesis and/or ovulation were also among the FOXL2-activated targets, including *Foxo1, Gja1* (a.k.a. *connexin 43*), *Ereg* (coding for the EGF-like Epiregulin), *Ptger2*, or *Has2*. Some FOXL2-repressed genes are involved in testis development, such as *Sox9, Dhh,* and *Sox8* (*Yao et al., 2002*; *Georg et al., 2012*). An analysis of Gene Ontology (GO) term enrichment showed that FOXL2 targets were characterized by keywords such as *cell adhesion* (p = 3.10⁻⁹), *regulation of cell proliferation* (p = 7.10⁻⁷), and *regulation of hormone levels* (p = 1.10⁻⁸) (*Table 1*). Other enriched terms were *reproductive structure development* (p = 3.10⁻⁵) and *developmental growth* (p = 2.10⁻⁵). The Panther Classification system showed that FOXL2 targets were enriched in genes involved in *cell communication* (p = 6.10⁻⁶) and *signal transduction* (p = 1.10⁻⁵). In particular, some of the most strongly activated targets are major players of important signal transduction pathways such as *Grem1* and *Grem2,* encoding BMP antagonists; *Rgs2* and *Rgs4,* encoding regulators of G-protein signaling, and the NRs *Esr2* and *Pparg*.

We compared the direct and indirect FOXL2 targets from our study with the genes modified by a twofold factor or more 3 weeks after the induction of *Foxl2* deletion in the conditional knockout mouse model of Uhlenhaut et al. (*Uhlenhaut et al., 2009*). Out of 345 FOXL2 target genes also present in the data set of Uhlenhaut et al., 71 were similarly affected following *Foxl2* deletion in the ovary (*Figure 1D* and *Figure 1—source data 1*). These notably included the critical factors for

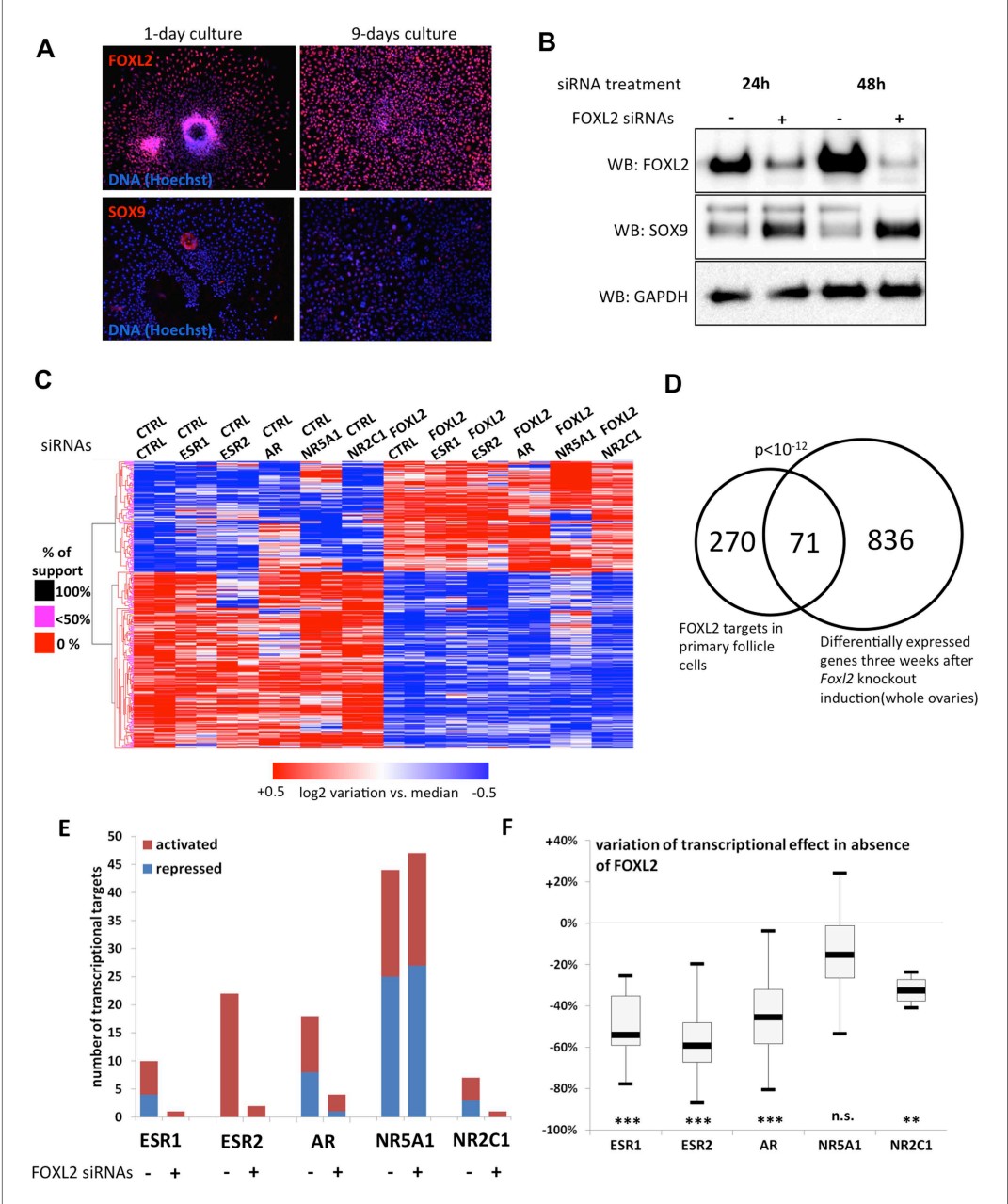

**Figure 1**. FOXL2 and nuclear receptor transcriptional targets in primary follicular cells. (**A**) Analysis by immunofluorescence of FOXL2 and SOX9 expression in primary murine granulosa cells cultured for either 1 day or 9 days. FOXL2 and SOX9 are stained in red, whereas DNA was counterstained with Hoechst 33342 (blue). These micrographs show that FOXL2 expression remains strong and homogenous in 9-day granulosa cell cultures, whereas SOX9 is slightly upregulated. (**B**) Western blot analysis of FOXL2 and SOX9 expression in 9-day cultured murine granulosa cells treated with a control siRNA or with anti-FOXL2 siRNAs for 24 or 48 hr. GAPDH was used as a loading control. This panel shows that FOXL2 is required for *Sox9* repression in these cells, as SOX9 expression increases quickly following *Foxl2* knockdown. (**C**) Heatmap representation of the relative expression values of FOXL2 targets in various conditions (biological duplicates). A bootstraped hierarchical clustering was performed and the dendrogram is represented on the left. The clustering discriminates with 100% of support two main groups of FOXL2-activated targets, and two groups of FOXL2-repressed targets (detailed below). (**D**) Venn Diagram representing the intersection of FOXL2 targets in our data with genes whose expression is modified 3 weeks after *Foxl2* conditional knockout in granulosa cells (in vivo). The intersection is composed of genes affected in the same direction in both studies. (**E**) Number of activated or repressed targets of the indicated NRs in the presence of
*Figure 1. Continued on next page*

*Figure 1. Continued*

FOXL2 (i.e., control siRNA) or in its absence (i.e., when comparing to conditions where anti-FOXL2 siRNAs were used). (**F**) Boxplot of the variation (in %) of the transcriptional effect of the indicated NRs on its target genes (defined according to the inclusion criteria explained in the main text), when *Foxl2* was knocked down compared to control conditions. This variation was calculated for each target gene of the indicated NR. The extreme points represent the 5th and 95th percentiles, whereas the box represents the 25th to 75th percentiles, with the median indicated in black. Statistical significance of the observed differences with the null hypothesis (reference value of 0) in a one-sample *t* test: n.s. non significant; **p < 0,01; ***p < 0,0001.

The following source data and figure supplements are available for figure 1:

**Source data 1**. This file contains all Log2 transcriptional effects and transcriptional targets analyzed in *Figure 1*, which were calculated as indicated in methods.

**Figure supplement 1**. Detection and quantification of various nuclear receptors in primary follicle cells.

**Figure supplement 2**. Characterization of siRNA efficiency for microarray analysis.

folliculogenesis *Fst*, *Cyp19a1*, *Esr2*, *Foxo1*, *Pparg*, *Gja1*, and *Bmpr1b*, which required FOXL2 for their expression, as well as the testis-determining or testis-specific factors *Sox9*, *Sox8*, and *Dhh,* which were repressed by FOXL2. FOXL2 ablation in granulosa cells results in a loss of their identity and in the appearance of testis-like structures, making difficult to evaluate which genes are first affected by the loss of FOXL2. These common targets may thus represent the early responders to FOXL2 deletion and play a critical role in the maintenance of granulosa cell identity, whereas other differentially expressed genes in the ovaries of mutant mice may be affected following the transdifferentiation of granulosa cells. GO term enrichment of the former genes reveals in particular their role in the *regulation of hormone levels* (p = 2.10$^{-5}$, *Table 1*). The intersection of both data sets provides strong candidates for in vivo FOXL2 targets in murine ovaries.

We then explored the transcriptional changes owing to the knockdowns of each of the five NRs. To ensure a good reliability of the potential targets obtained, we focused on genes whose expression displayed a twofold change or more (see 'Materials and methods' for details). We found that NR5A1, ESR2, and AR were the NRs had the highest numbers of potential direct and indirect targets (according to our inclusion criteria), whereas ESR1 and NR2C1 had very few targets (*Figure 1E* and *Figure 1—source data 1*). This suggests that the knockdown of either *Esr1* or *Nr2c1* may be compensated by their respective paralogs *Esr2* and *Nr2c2*, which are more strongly expressed in follicular cells (*Figure 1—figure supplement 1*). ESR2 only had activated targets, whereas AR and NR5A1 had as many activated as repressed targets.

We used the same strategy to explore how the absence of FOXL2 modified the expression of the targets of the NRs defined above. Interestingly, ESR2 and AR kept very few targets in the absence of FOXL2, whereas NR5A1 conserved a similar number of targets (*Figure 1E*). Consistently, the absence of FOXL2 led to halving the average transcriptional effect of ESR2 and AR on their target genes, whereas target modulation by NR5A1 underwent only a slight decrease (*Figure 1F*). This suggests that FOXL2 is required directly or indirectly for efficient target regulation by ESR2 and AR, but not by NR5A1.

## Genome-wide identification of potential regulatory elements recognized by FOXL2

To better understand how FOXL2 regulates its direct transcriptional targets, we performed a chromatin immunoprecipitation coupled to high-throughput sequencing (anti-FOXL2 ChIP-seq) in primary follicular cells. Using MACS as the peak detection algorithm, we detected about 14,000 peaks suggestive of FOXL2 binding to chromatin (*Figure 2—source data 1*). The analysis of these peaks revealed a highly enriched sequence corresponding to the characteristic motif of *forkhead* factors (5'-G/A-T-A-A-A-C/T-A-3'), with a very narrow enrichment window centered at the peak summits (*Figure 2A*). We analyzed the positions of FOXL2 peaks with respect to the annotated loci on the genome and found that transcription start sites (TSSs) were strongly enriched in FOXL2 peaks (*Figure 2—figure supplement 1A*). They were especially present within −300 bp and +100 bp around the TSSs (*Figure 2B*).

**Table 1.** Term enrichment among FOXL2 targets in cultured murine primary cells and FOXL2 targets shared with in vivo analysis

| ID | Term | Count | Bonferroni | p-Value | Genes |
|---|---|---|---|---|---|
| GO TERM ENRICHMENT of FOXL2 target genes | | | | | |
| GO:0007155 | cell adhesion | 34 | $6.10^{-6}$ | $3.10^{-9}$ | NRP2, CDK5R1, CLSTN2, MYBPC3, LMO7, CLDN11, SOX9, CD24A, CNTNAP5A, TGFB2, WISP2, WISP1, PVRL1, ADAM2, MSLN, ACAN, COL6A2, PSTPIP1, COL6A1, ITGA1, ITGA2, PCDH8, PCDH7, EMILIN2, MCAM, GPR98, PCDH18, AMIGO2, ITGA9, OMD, FREM2, ITGA7, VCAN, BMPR1B |
| GO:0022610 | biological adhesion | 34 | $6.10^{-6}$ | $3.10^{-9}$ | NRP2, CDK5R1, CLSTN2, MYBPC3, LMO7, CLDN11, SOX9, CD24A, CNTNAP5A, TGFB2, WISP2, WISP1, PVRL1, ADAM2, MSLN, ACAN, COL6A2, PSTPIP1, COL6A1, ITGA1, ITGA2, PCDH8, PCDH7, EMILIN2, MCAM, GPR98, PCDH18, AMIGO2, ITGA9, OMD, FREM2, ITGA7, VCAN, BMPR1B |
| GO:0010817 | regulation of hormone levels | 16 | $2.10^{-5}$ | $1.10^{-8}$ | HSD17B1, EXOC3L, ADH7, VGF, IRS1, ALDH1A2, ACE, RDH10, DIO2, AKR1C18, CHST8, HSD11B2, BMPR1B, PCSK5, SMPD3, CYP19A1 |
| GO:0042127 | regulation of cell proliferation | 29 | $1.10^{-3}$ | $7.10^{-7}$ | FGFR2, FGF18, PTGS2, PPARG, BTC, FOXO1, SOX9, CD24A, ADORA1, TGFB2, ALDH1A2, BDNF, SPRY1, S1PR1, CDKN2B, HEY2, GRPR, ADAM33, PTGER2, BMP2, ICOSL, IL7, EFNB1, ESR2, IRS1, EREG, EPGN, PLAU, BMPR1A |
| GO:0042445 | hormone metabolic process | 12 | $1.10^{-3}$ | $8.10^{-7}$ | ALDH1A2, ACE, RDH10, DIO2, AKR1C18, HSD17B1, CHST8, HSD11B2, ADH7, BMPR1B, PCSK5, CYP19A1 |
| GO:0040007 | growth | 16 | $5.10^{-3}$ | $3.10^{-6}$ | FGFR2, BMP2, PTGS2, GJA1, MREG, SOX9, TIMP3, TGFB2, RDH10, EREG, MFSD7B, ETNK2, BMPR1B, BMPR1A, CYP19A1, ADD1 |
| GO:0016337 | cell–cell adhesion | 17 | $1.10^{-2}$ | $8.10^{-6}$ | CDK5R1, CLSTN2, LMO7, PCDH8, CLDN11, PCDH7, MCAM, SOX9, CD24A, GPR98, TGFB2, PCDH18, AMIGO2, PVRL1, FREM2, ACAN, BMPR1B |
| GO:0001568 | blood vessel development | 17 | $2.10^{-2}$ | $1.10^{-5}$ | FGFR2, FGF18, TBX20, FOXO1, GJA1, CXCL12, TGFB2, ALDH1A2, S1PR1, EREG, EPGN, ITGA7, HEY2, ROBO4, ANGPT1, COL1A1, ANGPT2 |
| GO:0048589 | developmental growth | 11 | $3.10^{-2}$ | $2.10^{-5}$ | FGFR2, RDH10, EREG, PTGS2, GJA1, BMPR1B, MREG, SOX9, TIMP3, BMPR1A, CYP19A1 |
| GO:0001944 | vasculature development | 17 | $3.10^{-2}$ | $2.10^{-5}$ | FGFR2, FGF18, TBX20, FOXO1, GJA1, CXCL12, TGFB2, ALDH1A2, S1PR1, EREG, EPGN, ITGA7, HEY2, ROBO4, ANGPT1, COL1A1, ANGPT2 |
| GO:0048608 | reproductive structure development | 12 | $5.10^{-2}$ | $3.10^{-5}$ | FGFR2, DHH, RDH10, EREG, FST, ADAMTS1, ESR2, BMPR1B, VGF, SOX9, LFNG, CYP19A1 |

*Table 1. Continued on next page*

*Table 1. Continued*

| ID | Term | Count | Bonferroni | p-Value | Genes |
|---|---|---|---|---|---|
| PANTHER CLASSIFICATION of FOXL2 target genes | | | | | |
| BP00274 | Cell communication | 31 | $7.10^{-4}$ | $6.10^{-6}$ | CXCL5, CLSTN2, PPARG, BTC, FOXO1, CNTNAP5A, ASGR1, ASGR2, WISP2, BDNF, PVRL1, ADAM2, ANGPT1, HTR1D, ANGPT2, ODZ4, SLC8A1, IL7, EFEMP2, PCDH8, ESR2, PCDH7, GPR98, ARHGAP26, PCDH18, OMD, FREM2, EPGN, VCAN, PLAU, IGFBP4 |
| BP00102 | Signal transduction | 70 | $2.10^{-3}$ | $1.10^{-5}$ | NRP2, NDST4, CLSTN2, BTC, PPARG, GJA1, FOXO1, CNTNAP5A, ASGR1, ASGR2, BDNF, WISP2, S1PR1, GAB2, CXCR7, ADAM2, PDE4B, STARD8, ANGPT1, CHST15, HTR1D, RAMP1, ANGPT2, CCRL1, ODZ4, PTGER2, ARHGEF7, PCDH8, ESR2, PCDH7, IRS1, GPR98, ARHGAP26, AMIGO2, PLCE1, EREG, LYST, VCAN, RASD1, CAMK1D, FGFR2, IL1R1, CXCL5, SAV1, HS3ST1, PVRL1, GRPR, PCSK5, OLFM1, PIK3R1, SLC8A1, IL7, RASEF, EFEMP2, SPSB4, VMN2R11, SHANK2, RCAN2, PCDH18, DUSP5, EPHA4, OMD, RPS6KA1, RGS2, EPGN, FREM2, RGS4, BMPR1B, PLAU, IGFBP4 |
| GO TERM ENRICHMENT of the 'in vivo' FOXL2 target (71 genes at the intersection between our data and *Ulenhaut's 2009*) | | | | | |
| GO:0010817 | regulation of hormone levels | 7 | $1.10^{-2}$ | $2.10^{-5}$ | HSD17B1, CHST8, HSD11B2, EXOC3L, BMPR1B, PCSK5, CYP19A1 |
| GO:0042445 | hormone metabolic process | 6 | $3.10^{-2}$ | $4.10^{-5}$ | HSD17B1, CHST8, HSD11B2, BMPR1B, PCSK5, CYP19A1 |

Gene lists (FOXL2 targets or FOXL2 targets shared with *Uhlenhaut et al., 2009*) were analysed using the DAVID server (**david**.abcc.ncifcrf.gov/) to find significantly enriched terms among the GO Biological Process and Panther Biological Process databases. Only terms enriched with a Bonferroni p-value <0.05 were retained.

However, to our surprise, when we correlated the presence of FOXL2 peaks with the transcriptional effects of the *Foxl2* knockdown, we observed that the former were not over-represented near the TSSs (or the promoter regions) of its transcriptional targets (*Figure 2C*). However, we detected a strong enrichment of FOXL2 peaks in the regions containing the transcription units (TUs, extending from the TSS to the annotated polyA signal) of FOXL2-activated targets, but not in the TUs of the repressed ones (*Figure 2D*). This observation was even more striking considering only the 71 'in vivo' FOXL2 targets (*Figure 2—figure supplement 1B*). Since the TUs of protein coding genes are largely composed of introns (in terms of sequence length), this suggests that FOXL2 acts mainly through intronic regulatory elements (as previously described for *Fst*) (*Blount et al., 2009*). Indeed, a majority of FOXL2-activated targets had FOXL2 peaks in their introns (examples in the *Figure 2—figure supplement 2*), suggestive of the existence of potential regulatory elements recognized by FOXL2, although some target genes had no peak within their TUs (*Figure 2—figure supplement 3*). The enrichment of FOXL2 peaks in the TSSs may thus be a consequence of the decondensed chromatin state of these regions (i.e., nonfunctional binding). This observation is worth noting because the presence of ChIP peaks for transcription factors at TSSs is widely used as a criterion suggesting active transcriptional regulation (*Ouyang et al., 2009*), although as shown here for FOXL2, it may not always be the case. Interestingly, FOXL2-repressed genes were not enriched in FOXL2 peaks, in or at the proximity (50 kb) of their TUs. This notably included the testis determining genes *Sox9*, *Sox8*, and *Dhh*, suggesting that

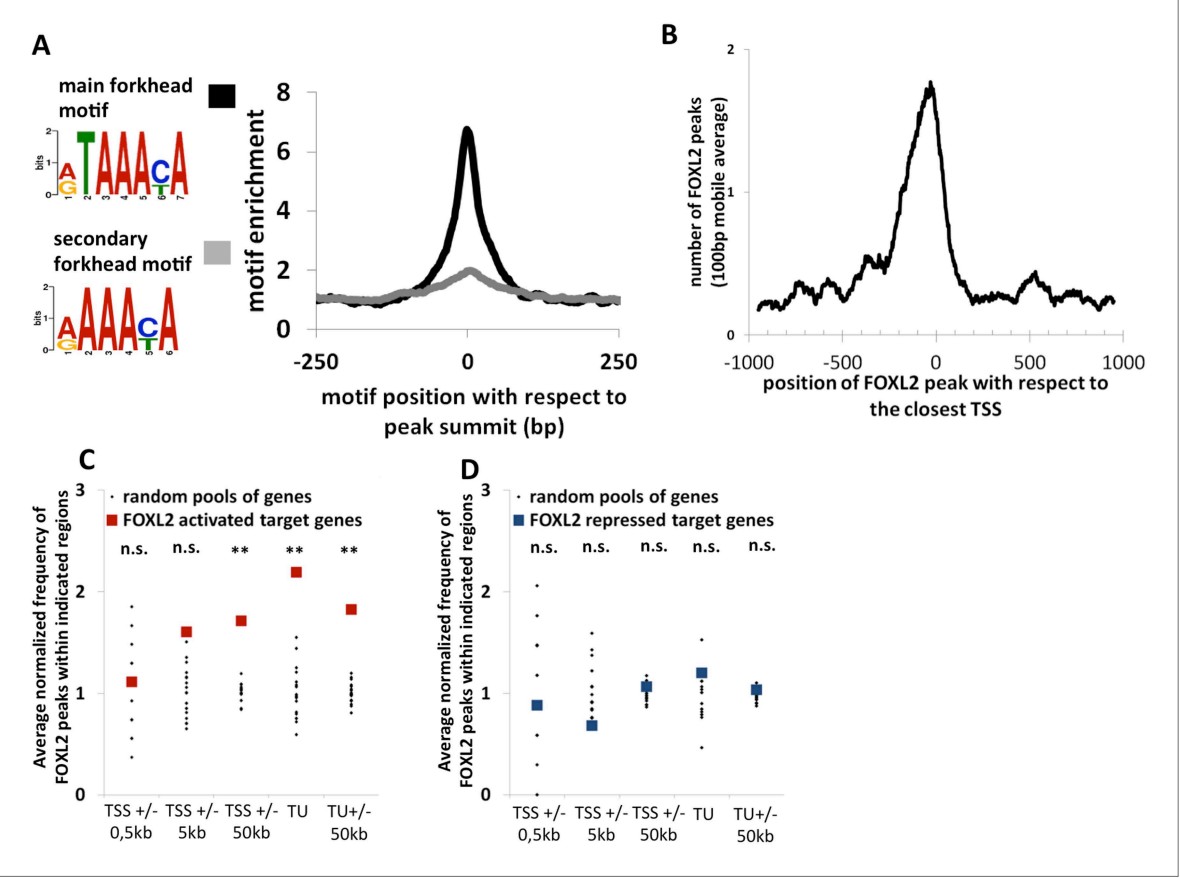

**Figure 2**. Genome-wide analysis of FOXL2 binding sites. (**A**) Logo representing two motifs over-represented in FOXL2 peaks and corresponding to forkhead motifs. The right panel represents the enrichment of these motifs in FOXL2 peaks, centered on their summits. (**B**) Distribution of FOXL2 peaks around TSSs. A stronger enrichment can be observed between −300 and +100 bp of the TSSs. (**C–D**) Enrichment of FOXL2 peaks at various locations of FOXL2-activated (**C**) and FOXL2-repressed (**D**) targets. Black dots represent the enrichment in FOXL2 peaks for 20 pools of randomly selected genes, each pool comprising the same number of genes than FOXL2-activated/repressed genes. The significance of FOXL2 peaks enrichment close to FOXL2 targets was evaluated using a Grubbs test for the detection of outliers. **$p < 0.01$; n.s.: non-significant. Surprisingly, FOXL2 peaks were not enriched around the TSSs of FOXL2 targets. Nevertheless, a strong enrichment of FOXL2 peaks can be observed in the transcription units (TUs) of FOXL2-activated targets compared to the TUs of randomly selected genes.

The following source data and figure supplements are available for figure 2:

**Source data 1**. MACS results providing the list of FOXL2-binding regions in mm9 canonical female genome coordinates.

**Figure supplement 1**. Further characterization of FOXL2 binding sites repartition compared to its transcriptomic targets.

**Figure supplement 2**. Alignment of the density of FOXL2-mapped reads and of FOXL2-detected peaks with the referenced transcription units of various gene activated by FOXL2.

**Figure supplement 3**. Alignment of the density of FOXL2-mapped reads and FOXL2-detected peaks with the referenced transcription units of various gene activated (A) or repressed (B) by FOXL2.

their repression by FOXL2 may be mostly indirect, at least in these cultured cells (*Figure 2—figure supplement 3*).

## FOXL2 directly regulates *Esr2* expression

As previously mentioned, we observed that *Esr2* expression dropped following *Foxl2* knockdown, showing that the latter is required for the expression of ESR2 in follicular cells. Previous studies have shown that *Esr2* expression is dependent on Activin/SMAD3 in granulosa cells (*Kipp et al., 2007*).

Since cooperation between FOXL2 and Activin/SMAD3 has already been identified on some targets, we tested whether FOXL2 was required for *Esr2* induction by Activin. For this, we treated primary follicular cells with siRNAs targeting FOXL2 or control siRNAs for 24 hr, and then with recombinant Activin A for 18 hr (*Figure 3A*). As expected, we observed an increase of *Esr2* expression in the presence of Activin A and a decrease following the *Foxl2* knockdown. Interestingly, no increase in *Esr2* expression could be observed in cells treated with Activin A when *Foxl2* was knocked-down. This clearly shows that FOXL2 is required for efficient Activin induction of *Esr2* expression.

The anti-FOXL2 ChIP-seq showed the presence of three main peaks in *Esr2* introns, which we named peaks 1 to 3 based on their distance from the TSS (*Figure 3B*). A sequence analysis of these elements showed that peak 3 contained a 20-bp DNA sequence composed of contiguous FOXL2, GATA, and SMAD elements (*Figure 3C*) and that peak 2 contained several potential FOXL2 elements. Conversely, peak 1 contained no obvious sequences corresponding to FOXL2 or SMAD3 motifs. To test whether these peaks could be regulatory elements, we cloned each of them upstream of a minimal SV40 promoter driving the expression of the luciferase gene. Next, we co-transfected these constructs with plasmids driving the expression of FOXL2 and SMAD3 in COV434 cells, human granulosa-derived cells, which are advantageous in this case as they do not express FOXL2 and very poorly express SMAD3. These cells were then treated with Activin A or a control solution (*Figure 3D*). We found that neither FOXL2 nor SMAD3 could activate the expression of luciferase through the peak 1 sequence, whereas only FOXL2 could activate luciferase expression through the peak 2 sequence. Interestingly, FOXL2 and SMAD3 synergistically activated luciferase expression through the peak 3 sequence (located in the 8th intron of *Esr2*), and this effect was enhanced when the cells were treated with Activin A. This observation was confirmed when we expressed a constitutively active Activin receptor (TGFBR1-T204D), suggesting that the observed effect is mediated by the canonical TGF-β pathway (*Figure 3E*). This is coherent with the presence of a FOXL2/SMAD3 composite element in the peak 3 sequence. We therefore created constructs where we mutated either the FOXL2 or the SMAD3 site in the composite element, and a construct in which the whole element was removed. As expected, we observed a strong reduction of luciferase expression when the SMAD3 or FOXL2 elements were mutated or when the whole composite element was removed (*Figure 3E*, 'full peak 3' series). However, a weaker synergistic activation of luciferase expression by FOXL2 and SMAD3 remained even in the absence of FOXL2 and SMAD3 elements, suggesting that other non-canonical DNA elements or interactions may account for part of FOXL2/SMAD3 synergy.

We observed that a canonical estrogen receptor-binding site was present nearby FOXL2 and SMAD3 sites in the peak 3 sequence. We therefore tested whether ESR2 could activate its own expression, alone or in combination with FOXL2 and TGFβ/SMAD3 (*Figure 3F*). Interestingly, we observed that the overexpression of ESR2 alone or in combination with SMAD3 and TGFBR1 (T204D) had little effect on luciferase activity. However, ESR2 was able to slightly induce luciferase in the presence of FOXL2 and more strongly in the presence of FOXL2, SMAD3, and TGFBR1 (T204D). Therefore, this experiment suggests that the induction of *Esr2* expression due to FOXL2 and TGFβ signals may be amplified by ESR2 action on its own enhancer, although ESR2 itself may not be able to stabilize alone its own expression. This further supports the idea that FOXL2 is required to establish a robust ESR2 expression in the ovary. Several observations corroborate this finding. Indeed, the transcriptional effect of *Foxl2* knockdown is well correlated with the one of ESR2 on its target genes (*Figure 4—figure supplement 1*). Thirteen out of the 20 ESR2 target genes detected here are also (direct and indirect) targets of FOXL2. These shared targets represent slightly more than 10% of FOXL2 targets, confirming that FOXL2 acts upstream of ESR2. In line with this, a hierarchical clustering of FOXL2-activated targets yields two main clusters (*Figure 4A*). The first cluster is mainly characterized by a strong effect of *Esr2* knockdown and includes ESR2 transcriptional targets, whereas no particular signature was observed for the second cluster (*Figure 4B* and *Figure 4—figure supplement 2*). These results show that gene regulation via, or along with, ESR2 may represent an important part of the transcriptional effects of FOXL2. Moreover, a third of the ESR2 targets in cultured cells, and *Esr2* itself, had a decreased expression in the *Foxl2* conditional-knockout ovaries, suggesting that FOXL2 regulates ESR2 (and its targets) in vivo (*Figure 1—source data 1*). Finally, it is interesting to note that although FOXL2 has a strong positive transcriptional effect on ESR2 targets in control conditions, it has in average no effect on the same genes when *Esr2* is knocked down (*Figure 4C*). FOXL2 still activates a few genes, such as *Fam84a*, although to lower extents (*Figure 4—source data 1*). This suggests that the

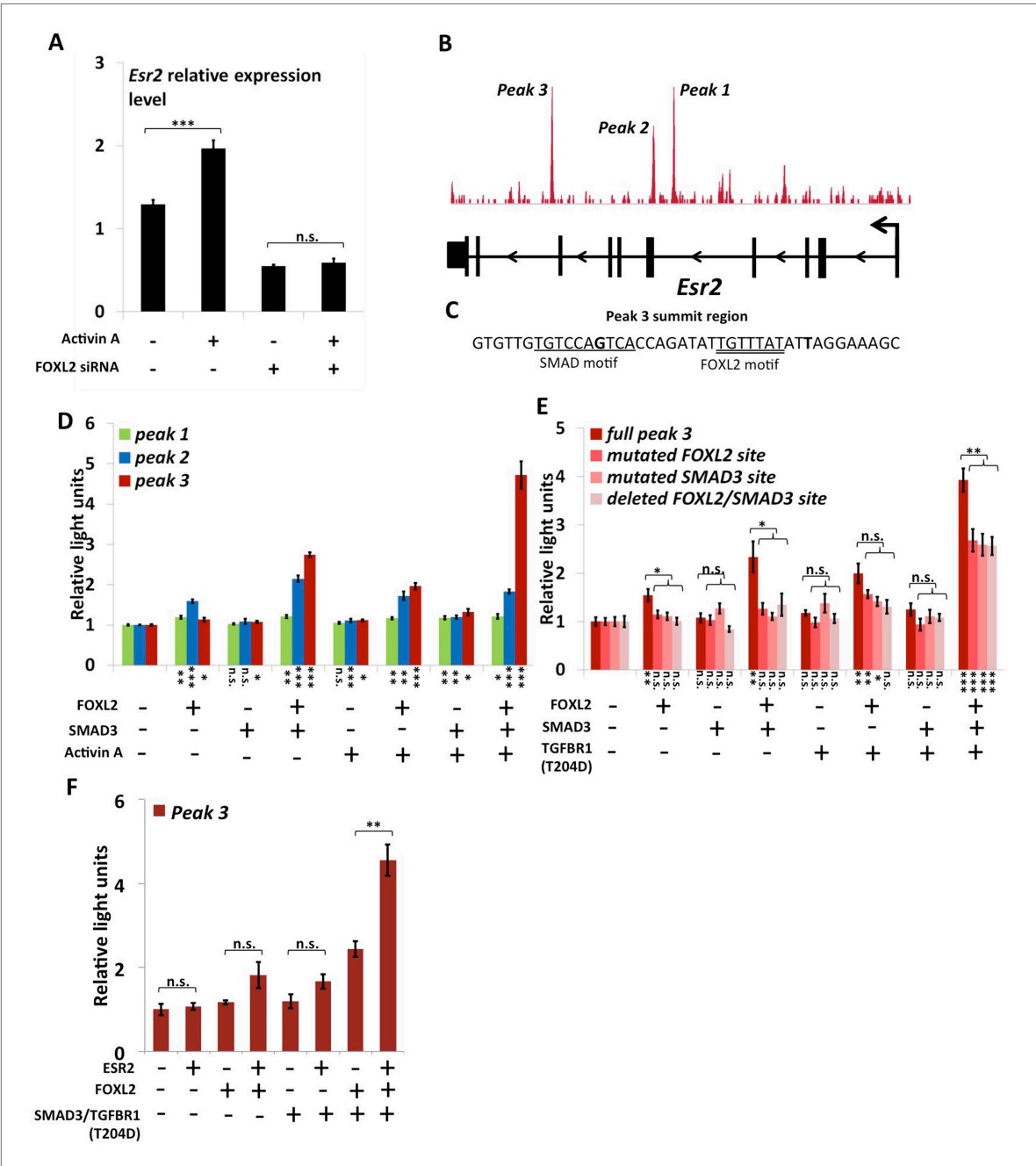

**Figure 3**. FOXL2 is required for ESR2 expression. (**A**) Relative amounts of *Esr2* cDNA, determined by qPCR, in mouse follicular cells transfected with an anti-FOXL2 or control siRNAs for 24 hr and treated by 2 nM Activin A or a vehicle for 18 hr. The average expression level of *Sdha* and *Actb* was used as a control for normalization. The values are the mean of three independent experiments performed in duplicate. (**B**) FOXL2 peaks in the *Esr2* TU. Three main peaks can be observed in the fourth and eighth introns. (**C**) Sequence around the summit of the FOXL2 peak in the eighth intron. FOXL2 and SMAD motifs are present near the peak summit. (**D–F**) Luciferase assays showing FOXL2/SMAD3/ESR2 transcriptional cooperation/synergy on the *Esr2* regulatory elements. COV434 cells were transfected with constructs driving the expression of FOXL2, SMAD3, ESR2, and/or a constitutive mutant of TGFBR1 (T204D), as well as with constructs containing the sequences of Peaks 1–3 (**D**), mutated versions of Peak 3 (**E**) or Peak 3 alone (**F**) cloned upstream of a minimal CMV promoter driving the expression of the Firefly luciferase gene, and a construct for the expression of the Renilla luciferase. In the indicated conditions, the cells were treated with 2 nM Activin for 18 hr before lysis. Transcriptional activity was measured as the ratio of Firefly/Renilla luciferases. p-values according to a Student's *t* test between the indicated conditions and the control condition: *p < 0.05; **p < 0.01; ***p < 0.001; n.s.: non-significant. Asterisks above the histogram in (**E**) represent the maximum p-value of a Student's *t* test between full Peak 3 transcriptional activity and all three mutants. Experiments were performed in six replicates.

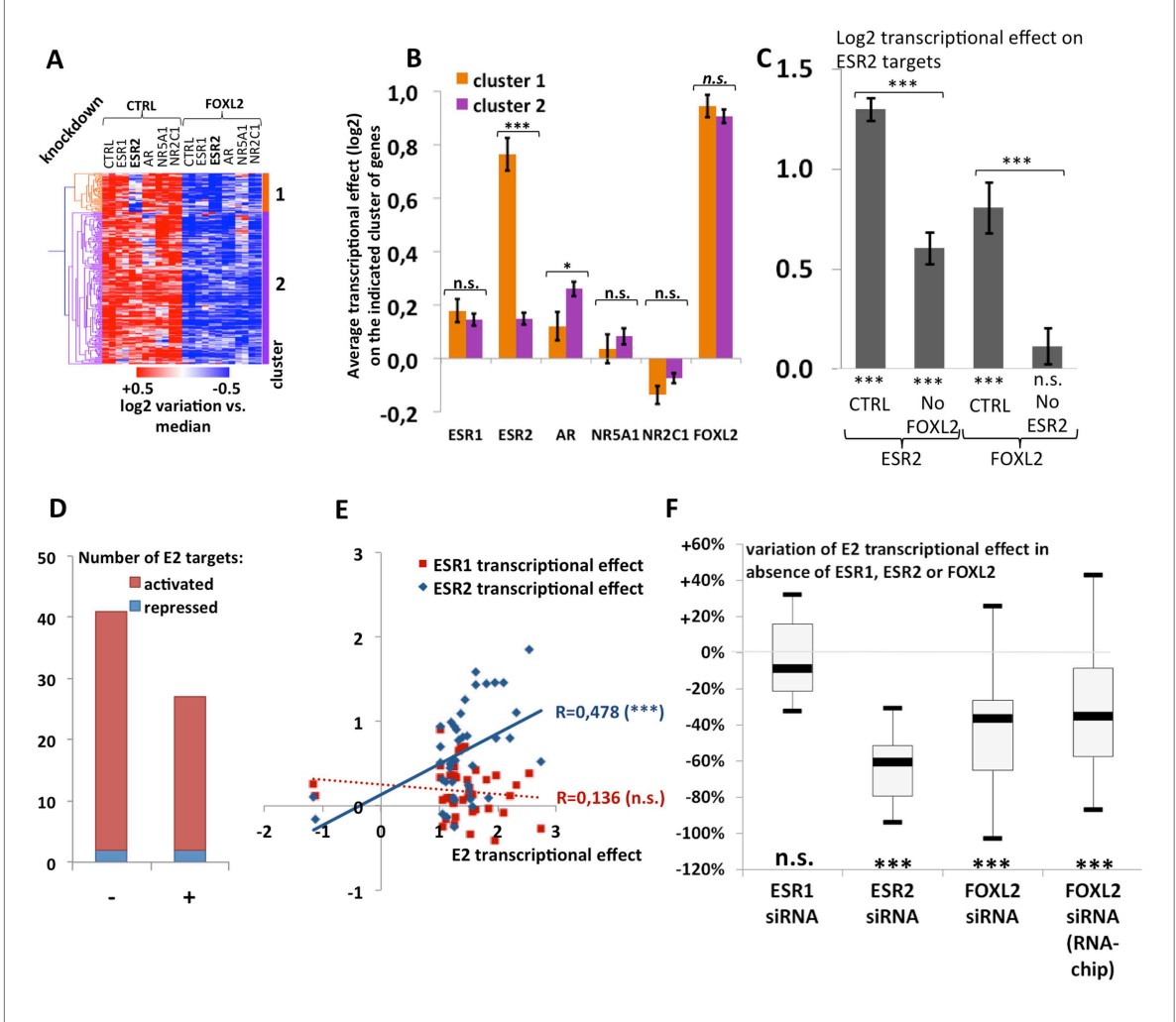

**Figure 4**. Analysis of estradiol-dependent transcription. (**A**) Heatmap of the FOXL2-activated target genes in our arrays. The targets were classified using hierarchical clustering. (**B**) Histogram of the average transcriptional effects of the five NRs and FOXL2 on the two clusters of FOXL2-activated target genes. The error bars represent the Standard Error of the Mean (SEM). Asterisks represent the p-value according to Student's *t* tests: *p < 0.05; **p < 0.01; ***p < 0.001; n.s.: non-significant. This diagram shows that the two clusters are mainly discriminated by the ESR2 effect, and that cluster 1 comprises mostly ESR2 transcriptional targets. (**C**) Histogram representing the average transcriptional effect of ESR2 (first two bars) and FOXL2 (last two bars) on the group of 20 ESR2 transcriptional target genes, either in normal conditions, or in the absence of FOXL2 (for ESR2) or ESR2 (for FOXL2). The diagram shows that ESR2 transcriptional effect on its targets is dramatically reduced in the absence of FOXL2. FOXL2 has a strong positive transcriptional effect on ESR2 targets, but this effect is abolished when *Esr2* is knocked-down. Asterisks above the graph represent the p-value according to two-sample Student's *t* tests: ***p < 0.001. Asterisks under the graph represent the p-value according to one-sample Student's *t* tests against the null hypothesis: ***p < 0.001. n.s.: p > 0.05. (**D**) Number of genes for which mRNA expression changed (2× or more) following a 10-hr E2 treatment in the presence of FOXL2 (i.e., in the presence of a control siRNA, left bar) or in the absence of FOXL2 (i.e., FOXL2-targeting siRNAs used for 24 hr before treatment, right bar). (**E**) Correlation of estradiol (E2) effect with that of either ESR1 (red squares) or ESR2 (blue diamonds) on E2-responding genes (log2-transformed data). Significance of the correlations according to a Fisher's F test: ***p < 0,001; n.s.: non-significant. (**F**) Boxplot of the variation of E2 transcriptional effect on E2 target genes (RNA-Chip lane) or 27 selected E2 target genes (first three lanes) between the control condition and the condition where *Foxl2, Esr1,* or *Esr2* is knocked down (as indicated). Statistical significance of the observed differences with respect to the reference value 0 in a one-sample *t* test: n.s. non significant, ***p < 0.001.

The following source data and figure supplements are available for figure 4:

**Source data 1**. This file contains all Log2 transcriptional effects and transcriptional targets analyzed in *Figure 4*, which were calculated as indicated in methods.

**Figure supplement 1**. Average transcriptional effect (log2) of each tested factor on ESR2 target genes.

*Figure 4. Continued on next page*

*Figure 4. Continued*

**Figure supplement 2**. Characterization of FOXL2 transcriptional targets.

**Figure supplement 3**. Characterization of estradiol transcriptional targets.

**Figure supplement 4**. RT-qPCR characterization of ESR1, ESR2, and FOXL2 siRNAs effects on E2 transcriptional induction.

**Figure supplement 5**. Synthetic representation of *Figure 4—figure supplement 4*.

overall positive effect of FOXL2 on ESR2 targets is indeed due to the activation by FOXL2 of ESR2 expression, although FOXL2 may also have some positive or negative effects on individual ESR2 targets through independent pathways.

## FOXL2 is required for efficient estrogen signaling

The above-mentioned findings support our hypothesis that FOXL2 plays an important role in estrogen signaling. We therefore analyzed the transcriptomic changes following a 10 hr 17β-estradiol (E2) treatment of primary granulosa cells pre-treated for 24 hr with either anti-FOXL2 or control siRNAs. We considered as potential targets those genes in the microarrays whose expression displayed a fold-change >2 in the presence of E2. We found 41 genes responding to estradiol according to our inclusion threshold, 39 of which were activated and only 2 were repressed. This is in agreement with previous reports suggesting that ERs are mainly transcriptional activators (*Kininis et al., 2007*) (*Figure 4D* and *Figure 4—source data 1*). The E2-activated targets comprised granulosa markers such as *Hsd11b2* (*Xu et al., 2011*), but also some genes involved in cancer such as *Tacc1, Angpt2,* or *Cbs* (*Ghayad et al., 2009*, 1; *Avraham et al., 2014*; *Weiner et al., 2012*). In agreement with previous works, E2 also repressed, at a lower level (<2-fold), the expression of *Sox9* and *Sox8*, involved in testis development.

We next compared the effect of E2 with the effect of *Esr1* and *Esr2* knockdowns on the same genes, as determined in our previous experiments. We found that the response to E2 was strongly correlated with the transcriptomic effect of *Esr2* knockdown, but not to that of *Esr1* (*Figure 4E*). This result suggests that ESR2, which is more expressed than ESR1 in these cells, is the main vector of estrogen signaling.

We then determined the number of genes affected by E2 treatment in the absence of FOXL2 and we observed a strong reduction of E2 effect in this case. Indeed, only 27 genes were E2 targets in the absence of FOXL2 (*Figure 4D*). In average, E2-dependent gene activation (in terms of mRNA levels) was reduced by about 35% in the absence of FOXL2 (*Figure 4F*). This supports the idea that FOXL2 is required for normal E2 signaling in follicular cells.

To have a broader view of the effect of FOXL2 on E2-induced transcriptional changes, we conducted an unsupervised hierarchical clustering of genes modified by >1.5-fold in the presence of E2 (*Figure 4—figure supplement 3A*). This analysis yielded three main groups of E2-activated genes and two groups of E2-repressed genes. These clusters were mainly characterized by the effect of the *Foxl2* knockdown (*Figure 4—figure supplement 3B*). However, the average effect of E2 in the presence or absence of FOXL2 was remarkably similar between the groups. Indeed, E2 effect was reduced as much (≈40%) for FOXL2-repressed genes than for FOXL2-activated ones. This suggests an indirect effect of FOXL2 on E2-dependent transcriptional changes, that is, that FOXL2 is required for the expression of a mediator of E2-dependent transcriptional regulation. However, a direct cooperation or antagonism between FOXL2 and ERs may occur on some genes.

Next, we selected 27 E2-activated genes among those having relatively high basal expression levels and tested by qPCR whether E2 could elicit their activation in the absence of ESR1, ESR2, or FOXL2 (*Figure 4F* and *Figures 4—figure supplement 4 and 5*). Not surprisingly, E2 activation was mostly maintained in the absence of ESR1, but it was strongly decreased upon *Esr2* knockdown. This is coherent with the correlation of the effect of E2 with only the effect of ESR2 and confirms that ESR2 is the main effector of estradiol signaling in these primary cells. The same experiment in the presence of anti-FOXL2 siRNAs showed a 40% reduction of E2-dependent gene activation, similar to what was observed on microarrays (*Figure 4F*). Altogether, these results show that FOXL2, mainly through its

regulation of *Esr2* expression, is required for the establishment of normal estradiol signaling in follicular cells.

## A feed-forward loop reinforces the impact of FOXL2 on granulosa cell identity

To better understand the combined effect of FOXL2 and ESR2 on granulosa cell maintenance, we analyzed SOX9 expression at the protein level 48 hr after *Foxl2* and/or *Esr2* knockdown. As expected considering our transcriptomic analyses, we observed that SOX9 expression was upregulated in the absence of either FOXL2 or ESR2 (*Figure 5A*). SOX9 expression was much higher in the absence of both ESR2 and FOXL2. This confirms that estrogen receptors and FOXL2 co-regulate SOX9 expression (*Uhlenhaut et al., 2009*). Interestingly, our transcriptomic analyses indicate that FOXL2 is required for the expression of *Cyp19a1* (encoding the rate-limiting enzyme in estradiol production) both in the presence (*Figure 5B*) and in the absence of hormones (*Figure 5C*). Thus, FOXL2 may allow both estradiol production and enable estrogen signaling in the same cells. These data suggest the existence of a coherent feed-forward loop in which FOXL2 stimulates both estradiol production and receptivity (i.e., ESR2 expression) that might be responsible at least in part for the maintenance of granulosa cell identity by repressing the testis determinant Sox9.

As previous work showed that FOXL2 and ESR1 were both required for SOX9 repression in the ovary, we tested whether E2 and ESR1 could have an influence on SOX9 expression in our system. We therefore deprived follicular cells of hormones for 24 hr before treating them with siRNA pools for 24 hr, and finally with E2 for 10 hr before lysis. We observed that E2 treatment induced a repression of SOX9 expression, in coherence with our transcriptomic analysis (*Figure 5D*). *Esr1* knockdown, on the other hand, had no clear effect on SOX9 expression or on the efficiency of SOX9 repression following E2 treatment. However, we observed that ESR2 knockdown resulted in an increase of SOX9 expression even in the virtual absence of hormones and that E2-induced repression was reduced in the absence of ESR2. Finally, SOX9 expression was much increased in the absence of FOXL2, but E2 was still efficient in repressing *SOX9* in this case. Altogether, our data show that FOXL2 and ESR2/E2 repress somewhat independently SOX9 expression in follicular cells. It has previously been suggested that FOXL2 and ESR1 bind together to a Testis-specific enhancer of *Sox9* to repress its expression. Here, we have detected no binding of FOXL2 to this sequence or to any other close sequence in our ChIP-Seq analysis. It is possible that the binding was lost during cell culture or that it occurs at a different stage of follicular development. However, the fact that FOXL2/E2 repression of SOX9 is still operant in our case suggests that several pathways orchestrated by FOXL2 cooperate to repress SOX9 expression in these cells (*Figure 5E*).

## Discussion

Our results provide a considerable number of direct and indirect targets of FOXL2 and of various NRs in follicular cells, but shows above all how FOXL2 may establish estrogen signaling in the ovary. Indeed, we show that FOXL2 is essential for Activin-dependent up-regulation of *Esr2* in follicular cells. This involves synergistic interactions between FOXL2 and SMAD3 on a regulatory element located within the 8th intron of the *Esr2* gene. Furthermore, we were able to show that ESR2 is the main vector of estradiol signaling in cells from preantral follicles, which is coherent with the fact that *Esr2* is much more expressed than *Esr1* in such growing follicles. Previous studies also suggested that ESR1 is poorly expressed in granulosa cells of growing follicles whereas ESR2 was often described as the main estrogen receptor in these cells (*Lenie and Smitz, 2008*). We provide evidence that FOXL2 has a major indirect impact on estradiol signaling through its direct regulation of *Esr2*. This role of FOXL2 on ESR2 signaling is also likely to exist in vivo (i.e., in the ovary) and expands its already known role in estrogen production through the regulation of aromatase expression, confirmed here. This suggests the existence of a coherent feed-forward loop in which FOXL2 stimulates both E2 production and receptivity (*Figure 5*) to maintain granulosa cell identity, as FOXL2, ESR2 and E2 all participate in SOX9 repression. This model is complementary of a previously proposed mechanism in which FOXL2 and ESR1 directly cooperated to repress *Sox9* expression though the TESCO enhancer (*Uhlenhaut et al., 2009*). We could not detect any binding of FOXL2 nor ESR1 to this region in our cells (data not shown), whereas the ability of FOXL2 and estrogens to repress *Sox9* was still present. Moreover, this repression acted both through ESR2/E2 and independently of estrogens. This shows that FOXL2 represses *Sox9* and maintains ovarian identity through multiple pathways, some of them yet to be

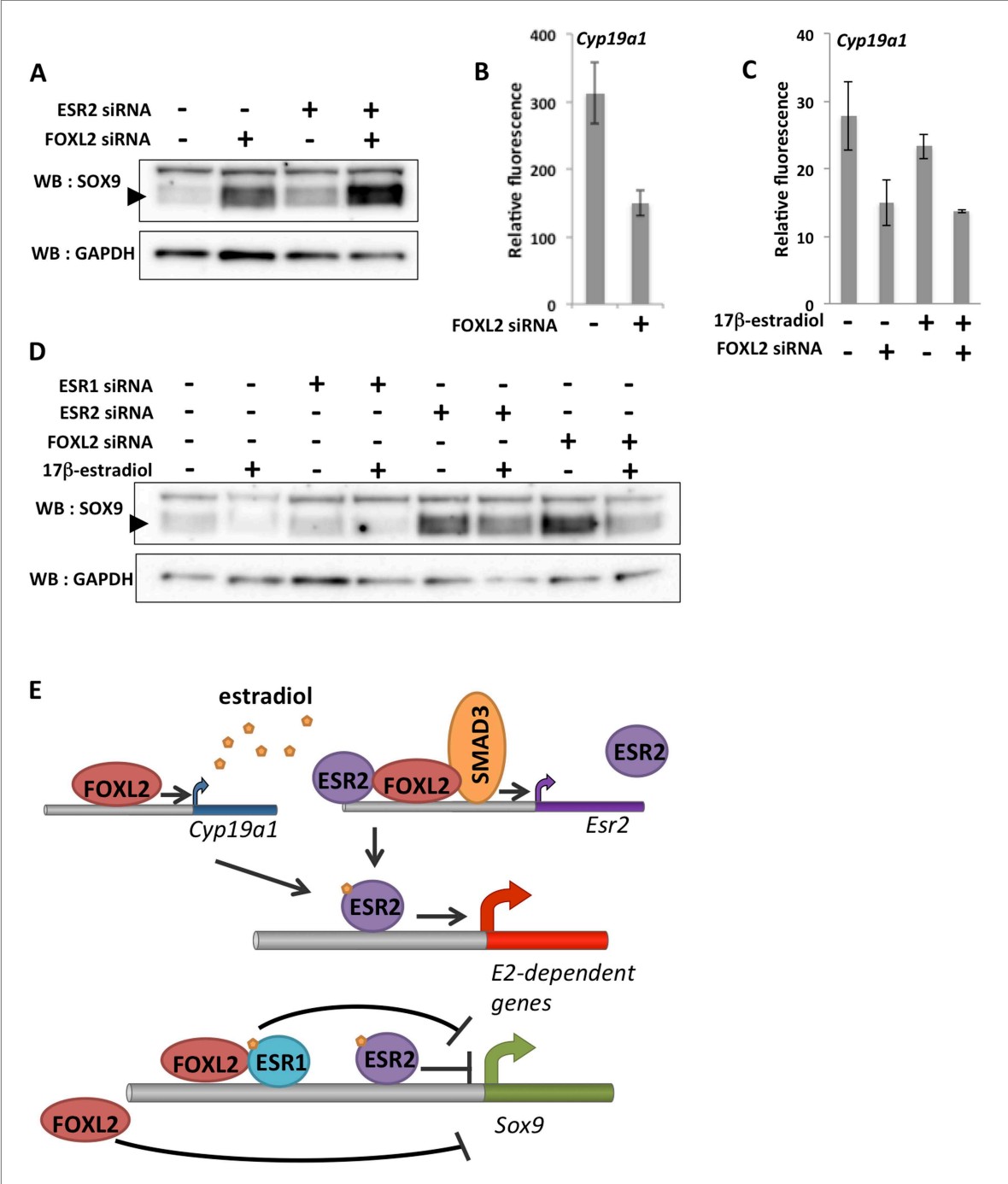

**Figure 5**. A robust network repressing SOX9 expression and promoting estradiol signaling. (**A**) Western blot analysis of SOX9 and GAPDH expression in the extracts of follicle cells treated with anti-FOXL2 and/or anti-ESR2 siRNA pools for 48 hr. A black arrow designates SOX9. (**B–C**) Histogram representation of *Cyp19a1* expression determined by our microarray analyses. Error bar represents the standard deviation based on two biological replicates. (**D**) Western blot analysis of SOX9 and GAPDH expression in extracts of hormone-deprived follicular cells treated with anti-FOXL2, anti-ESR1 or anti-ESR2 siRNA pools for 24hr and/or 17β-estradiol for 10 hr. (**E**) Schematic representation of our findings on FOXL2 regulation of estradiol signaling. Our findings suggest that FOXL2 may regulate estradiol signaling through (i) regulation of the production of E2 and (ii) regulation of estrogen receptivity by modulating the expression of *Esr2*. FOXL2 cooperates with Activin/SMAD3 and possibly ESR2 itself in the latter process. Sox9 repression seems to rely on multiple pathways in addition to the one described here, as FOXL2 and ESR1 have been described to co-bind a SOX9 regulatory element in the ovary.

discovered. Our results are fully coherent with a recent study showing that *Foxl2* deletion almost fully abolishes testis feminization due to *Dmrt1* deletion, while *Esr2* deletion only partially affects feminization (**Minkina et al. 2014**). Further work will be necessary to fully characterize this repression and to identify other signaling pathways implicated in granulosa cell differentiation. We are aware that our in vitro system is sub-optimal and that follicular cells can undergo some degree of trans-differentiation over the course of the culture period. Thus, our results will benefit from validation in vivo as methods to work with small amounts of material improve in the future.

It is worth noting that FOXL2 regulates the expression of many factors involved in several signaling pathways of utmost importance in the ovary, such as TGF-β/BMP signaling factors (*Fst, Bmp2, Bmpr1b*), the MAP-kinase cascade (*Adora1, Map3k5, Gadd45b*), steroid signaling (*Cyp19a1, Hsd11b2, Hsd17b1, Esr2*), PPAR signaling (*Pparg, Slc27a6, Fabp3*), or PI3K/Akt signaling (*Foxo1, Angpt1, Angpt2)*, which mediates part of FSH action in the ovary. Disruption of each of these pathways in the ovary is a cause of POI and their malfunction is commonly associated with oncogenesis and cancer progression. In this context, our finding that FOXL2 is required of proper estradiol signaling in the ovary may thus be an initial step towards a better molecular appraisal of the role of FOXL2 in global granulosa cell signaling response and its implication in tumorigenesis.

## Materials and methods

### Reagents and constructs

The rabbit anti-FOXL2 antibody has previously been described. The rabbit anti-SOX9 antibody was a kind gift of Dr Brigitte Boizet. The mouse anti-GAPDH antibody was purchased from ABM (ref G041) and the anti ER-a (HC-20) antibody was purchased from Santa Cruz Biotechnology (SCBT, Dallas, Texas) at a concentration of 2 µg/µl (ref SC-543X).

The full-length coding sequence of *Esr2* was amplified from primary follicular cells cDNA and cloned in frame with a V5 tag using the pcDNA3.1 TOPO-TA/V5-His cloning kit (Life Technologies, Carlsbad, California) according to manufacturer's instruction. FOXL2 and ESR1 potential regulatory elements identified from ChIP-Seq were amplified from primary follicle cell genomic DNA and inserted between the KpnI and NheI sites of the pGL3-promoter vector (Promega, Madison, Wisconsin). Vectors driving the expression of FOXL2, SMAD3 have been described previously (**Anttonen et al., 2014**), and the vector expressing constitutively active TGFBR1 (with T204D mutation) was a kind gift of Dr Peter Scheiffele (**Gore et al., 2005**).

### Primary follicular cells preparation and culture

The ovaries of 8-week-old female Swiss mice (pools of 5 mice per replicate for transcriptome analysis, 20 mice per condition for ChIP experiments) were dissected and digested in DMEM/F-12 medium (Life Technologies) containing 1 mg/ml collagenase (Sigma–Aldrich, Saint-Louis, Missouri), 0.5 mg/ml BSA (Sidma–Aldrich), and 2000 U/ml DNase I (Thermo-Fisher Scientific), in a ratio of 500 µl per ovary, at 37°C for 30 min. Ovaries were progressively dispersed during digestion using pipette tips of decreasing size to flush the organ debris. The mix was then diluted in 10 vol of DMEM/F-12 medium, filtered using a 500-µm membrane, spun at 300×*g*. The pellet was rinsed in cold PBS and spun again, and the pelleted debris was resuspended in a PBS/Ficoll mix of density 1.10. A PBS/Ficoll gradient was made by depositing PBS/Ficoll layers of density 1.08, 1.07, 1.06, 1.05, 1.03, and 1. The gradient was spun at 1500×*g* for 20 min, and 1 ml fractions were collected and diluted in 5-ml culture medium (i.e., DMEM/F-12 supplemented with 4% FBS, 1% Insulin-Transferrin-Selenium-X supplement, 1% Penicillin-Streptomycin [Life Technologies], and $10^{-7}$M androstenedione (Sigma–Aldrich). We then manually collected intact preantral follicles, characterized by the presence of several layers of granulosa cells around the oocyte and the absence of antrum. They mainly migrated to the 1.07/1.06 and 1.06/1.05 interfaces. Follicles were then rinsed, transferred to culture plates, and grown overnight. On day 1, they were dispersed using trypsin and then cultured in monolayers. Culture medium was changed on day 3, and cells were passed on days 5 and 7. For estradiol induction experiments, we used Phenol Red-free DMEM/F-12 and Optimem, charcoal-treated FBS (Life Technologies), and no androstenedione was added to culture medium.

### RNAi transfection

After 7 days of culture, cells were seeded for transfection in 12-well plates. After an 18-hr incubation, cell medium was replaced with a premade Mix containing 500 µl Optimem (Life Technologies), a total

of 12 pmol of siRNAs (2 × 6 pmol in the case of simultaneous knockdown), 200 ng pDsRed plasmid (Clontech Laboratories, Mountain View, California), and 2 µl Lipofectamine 2000 (Life Technologies). 6 hr after transfection, 1.5 ml of culture medium was added to each well, and cells were then grown for 24 hr before lysis in 500 µl Trizol reagent (Life Technologies), for a total culture time of 9 days.

The ON-TARGETPlus Smartpool siRNAs used here were purchased from Thermo Fisher Scientific (now distributed under the Dharmacon brand by GE Healthcare, Little Chalfont, Buckinghamshire, United Kingdom): non targeting pool (ref D-001810-10-05), Mouse anti-FOXL2 siRNAs (L-043309-01-0005), mouse ESR1 (L-058688-01-0005), mouse ESR2 (L-065564-01-0005), mouse AR (L-050296-00-0005), mouse NR5A1 (L-051262-01-0005), mouse NR2C1 (L-061593-01-0005). RNA/protein preparations were performed according to manufacturer's instruction.

## Transcriptome analyses

For the first experiment, 100 ng total RNA was reverse-transcribed and linearly amplified using the Whole Transcriptome Amplification Transplex kit (Sigma–Aldrich), according to the manufacturer's instructions. The amplified cDNAs from the 12 samples were sent to the Nimblegen expression platform (Roche NimbleGen, Reykjavík, Iceland) in biological duplicates (independent mice pools, primary cultures, amplifications, and hybridizations per RNA sample). Samples were hybridized on two high-density *Mus musculus* MM9 Gene Expression 12 × 135K array set, each one allowing the study of 12 samples in parallel and interrogating over 45K annotated transcripts of the Mm9 genome assembly. Nimblegen performed sample labeling, hybridization, and fluorescence acquisition and provided us with normalized data files. Data were deposited in Gene Expression Omnibus (Series GSE60764).

For the second experiment, 50 ng total RNA was amplified and labeled with Cy3 using Low input Quick Amp labeling Kit (Agilent technologies, Santa Clara, California) according to manufacturer's instruction. Labeled cRNA was then fragmented and hybridized over 1 Sureprint G3 Mm9 8 × 60K array (Agilent technologies), allowing the study of 8 samples in parallel and interrogating over 40K annotated transcripts of Mm9 assembly. Hybridization was performed according to manufacturer's instructions. Raw fluorescence data were extracted using Genespring software v12 (Agilent technologies) and normalized using the 75th centile method in each array. To allow for an easy comparison between the two experiments, we then took for each gene the average expression value of all transcripts. We removed from further analysis the genes for which median fluorescence was under 100 (NimbleGen data) in the first experiment, under 10 in the second experiment (Agilent data). Values were then log2 normalized and subtracted of the median value for each replicate. Data were deposited in Gene Expression Omnibus (Series GSE60770).

To uncover FOXL2 targets in the first experiment, we conducted a paired Significance Analysis for Microarrays using the Multiexperiment Viewer program v4.9 (http://www.tm4.org/mev.html). Each array treated with a control siRNA pool was paired with the corresponding array treated with a FOXL2 targeting siRNA pool, giving 12 independent pairs of arrays. Delta value was adjusted to the minimum value allowing a median FDR = 0. The differentially expressed genes were then sorted according to the average absolute difference between the control siRNA/control siRNA and control siRNA/FOXL2 siRNA conditions, and all genes above a threshold of 0.584 (i.e., log2(1.5)) were selected as FOXL2 potential targets.

To infer NR targets in the first experiment (NimbleGen data) and E2 targets in the second experiment (Agilent data), we sorted genes according to the average absolute difference between the control siRNA/control siRNA and control siRNA/NR siRNA conditions (or control siRNA/E2 treatment condition). All genes above a threshold of 1 (i.e., log2(2)) were retained. We then verified for each gene that this difference was superior to twice the sum of the standard deviations for each condition. The retained genes were considered as NRs or E2 potential targets.

## Chromatin immunoprecipitation

After 9 days of culture, cells grown to confluence in three 150-mm plates were fixed in situ with 1% formaldehyde for 10 min at room temperature, fixation was then blocked with 125 mM glycine, cells were rinsed and scrapped in PBS. Cells were first lysed in Lysis buffer A (50 mM Hepes pH 7.5, NaCl 140 mM, EDTA 1 mM, glycerol 10% NP-40 0.5%, and Triton X-100 0.25%), then spun and rinsed in lysis buffer B (Tris 10 mM pH 7.5, NaCl 200 mM, EDTA 1 mM, EGTA 0.5 mM), centrifugated again, and finally resuspended in lysis buffer C (Tris 10 mM pH 7.5, NaCl 100 mM, EDTA 1 mM, EGTA 0.5 mM, Na–deoxycholate 0.5%, N-lauryl sarcosine 0.5%). Chromatin was sheared in a Bioruptor sonicator,

used at high power with 30/30 s cycles for 20 min, then 1% Triton X-100 was added to lysates, and cell debris were centrifuged. In parallel, 30 µg anti-FOXL2 or anti-ESR1 was incubated with 200 µl ProteinG Dynabeads for 4 hr in PBS containing 0.1% BSA. Lysates were incubated with the beads overnight, beads were then washed five times in wash buffer (50 mM Hepes pH 7.6, 500 mM LiCl, 1 mM EDTA, 0.7% Na–deoxycholate, 1% NP-40) and once in TBS. DNA–protein complexes were then eluted in 10 mM Tris pH 8, 1 mM EDTA, 1% SDS at 65°C for 15 min. Cross links were reversed at 65°C overnight, then recovered DNA was treated with RNaseA, proteinase K, and purified using phenol–chloroform. Input samples underwent to the same procedure.

ChIP-Seq was performed by the Platform IMAGIF from the french Centre National de la Recherche Scientifique (www.imagif.cnrs.fr) on an Illumina HiSeq1000 instrument using standard protocols (single reads). The reads in fastq files were imported on Galaxy server (https://usegalaxy.org/). Reads were mapped on mm9 Canonical Female genome using Map Bowtie for Illumina tool with commonly used settings. Unmapped reads were then removed using the Filter SAM tool. Peak calling was performed using MACS tool using the following settings: Tag size: 50; Band width: 160 (first experiment) or 174 (second experiment); Mfold: 20; new peak detection method: yes; and other settings unchanged. Input DNA analyzed in parallel was used as a control for peak detection. Peaks with an FDR <0.05 were kept for further analysis. De novo motif discovery was conducted using the MEME-chip pipeline on MEME server (http://meme.nbcr.net/meme/). Data were deposited in Gene Expression Omnibus (Series GSE60858).

### Luciferase assays

COV434 cells were seeded 16 hr prior to transfection at a density of $4.10^4$ cells/cm$^2$ and transfected using the calcium–phosphate method. The biological activity of luciferase reporter constructs (see mains text) was assessed by the Dual-Luciferase Reporter Assay System (Promega). Relative luciferase units represent the ratio of firefly luciferase activity over Renilla luciferase activity in the samples. Each value is the mean of six independent experiments, and standard error bar represent the standard error of the mean (SEM).

## Acknowledgements

This work was supported by Centre National de la Recherche Scientifique (CNRS), La Ligue Nationale contre le Cancer and l'Université Paris Diderot-Paris7. We thank Claude Thermes, Delphine Naquin, Erwin van Dijck, Maud Silvain, and Yan Jaszczyszyn from the CNRS' IMAGIF platform for their most kindful help. Animal studies were conducted according to the guidelines of the Animal Experimentation Ethical Committee Buffon (CEEA-40), project CEB-25-2012. All high-throughput data are available at the Gene Expression Omnibus database and can be accessed through the SuperSeries GSE60859.

## Additional information

### Funding

| Funder | Author |
| --- | --- |
| Ligue Contre le Cancer | Reiner A Veitia |
| Agence Nationale de la Recherche | Reiner A Veitia |
| Universite Paris VII | Alain Zider, Reiner A Veitia, David L'Hôte, Anne Laure Todeschini, Aurélie Auguste, Bérengère Legois |
| Centre National de la Recherche Scientifique | Adrien Georges, David L'Hôte, Anne Laure Todeschini, Aurélie Auguste, Bérengère Legois, Alain Zider, Reiner A Veitia |

The funders had no role in study design, data collection and interpretation, or the decision to submit the work for publication.

### Author contributions

AG, DL'H, ALT, RAV, Conception and design, Acquisition of data, Analysis and interpretation of data, Drafting or revising the article; AA, Acquisition of data, Analysis and interpretation of data, Drafting

or revising the article; BL, Acquisition of data; AZ, Analysis and interpretation of data, Drafting or revising the article

### Ethics

Animal experimentation: This study was performed in strict accordance with the recommendations in the Guide for the Care and Use of Laboratory Animals of the Centre National pour la Recherche Scientifique and the Université Paris Diderot/Paris VII. Animal studies were conducted along the guidelines of the Animal Experimentation Ethical Committee Buffon (CEEA-40), project CEB-25-2012.

## Additional files

### Major dataset

The following dataset was generated:

| Author(s) | Year | Dataset title | Dataset ID and/or URL | Database, license, and accessibility information |
|---|---|---|---|---|
| Adrien Georges, David L'Hôte, Anne Laure Todeschini, Aurélie Auguste, Bérangère Legois, Alain Zider, Reiner A Veitia | 2014 | Data from: The transcription factor FOXL2 mobilizes estrogen signaling to maintain the identity of ovarian granulosa cells | GSE60859; http://www.ncbi.nlm.nih.gov/geo/query/acc.cgi?acc=GSE60859 | Publicly available at GEO (http://www.ncbi.nlm.nih.gov/geo/). |

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
