## [Decision Letter]

Thank you for sending your work entitled "The transcription factor FOXL2 mobilizes estrogen signaling in the maintenance of ovarian granulosa cells identity" for consideration at *eLife*. Your article has been favorably evaluated by Janet Rossant (Senior editor) and 3 reviewers.

The Senior editor and the three reviewers discussed their comments before we reached this decision, and the Senior editor has assembled the following comments to help you prepare a revised submission.

This is a well thought-out, well designed study investigating the role of FOXL2 in regulating the normal biology of ovarian granulosa cells. By using primary granulosa cell cultures, the authors have overcome some of the limitations of cell numbers involved in studying *in vivo* ovarian function. Consistent with findings from previous publications, the authors have shown that FOXL2 activates genes that are crucial for normal granulosa cell function (folliculogenesis and ovulation), and represses genes that are involved in testis development (male-determining transcription factor SOX9) and thus the maintenance of granulosa cell identity. Additionally, the results from ChIP-seq have provided further insights on how FOXL2 regulate its transcriptional targets. By correlating the ChIP-seq data with the gene expression knockdown data, the authors have demonstrated that FOXL2 acts predominantly through intronic regulatory elements, particularly for FOXL2-activated targets. Moreover, the authors were also able to show that FOXL2 directly modulates ESR2 expression via a previously uncharacterized intronic element to regulate estrogen signaling in granulosa cells. Lastly, the authors proposed a feed-forward mechanism by which FOXL2 increases estrogen production (through CYP19A gene induction) and activates ESR2 expression to reinforce its repressive effects on SOX9 expression. This manuscript is a nice addition to the current literature on FOXL2 and its role in normal ovarian function.

The reviewers did raise some concerns that need to be addressed before we can consider the manuscript for publication.

1) Use of an *in vitro* system is sub-obtimal because of the potential loss of cell identity over time. First, it is reasonable to question whether this *in vitro* system is a good reflection of the *in vivo* cell environment. Although the authors show that the explanted cells can maintain FOXL2 expression (and repress SOX9 expression) for 9 days in culture, we should be cautious about how cell culture affects all aspects of the study, as loss of granulosa identity is likely a gradual process beginning soon after explant. The time between cell explant and transcriptome/ChIP measurements is not clear in the methods.

2) Another concern is that these investigators did not detect binding of FOXL2 (or ESR1) to the TESCO element upstream of SOX9 which has been mapped *in vivo* in adult granulosa cells. This suggests that this study may not accurately reflect FOXL2 binding sites *in vivo* in all cases.

3) The study derives several key conclusions based upon knockdown experiments. The manuscript, however, does not provide some important experimental details and information on controls. It is evident from the data in Figure 1—figure supplement 2 that specific and selective knockdown has been achieved by each of the siRNAs. The author, however, fail to explain what is used as "control". Ideally, the control cells should express non-specific siRNA and effects of a control siRNA should be validated. In the legend for Figure 1—figure supplement 2 the authors state the following: "levels were normalized by the expression levels of Actb and Sdha". The authors should choose one internal control and specify which was used.

---

## [Author Response]

*1) Use of an in vitro system is sub-obtimal because of the potential loss of cell identity over time. First, it is reasonable to question whether this in vitro system is a good reflection of the in vivo cell environment. Although the authors show that the explanted cells can maintain FOXL2 expression (and repress SOX9 expression) for 9 days in culture, we should be cautious about how cell culture affects all aspects of the study, as loss of granulosa identity is likely a gradual process beginning soon after explant. The time between cell explant and transcriptome/ChIP measurements is not clear in the methods*.

Our *in vitro* system might indeed be considered as sub-optimal. However, this is the best solution we could implement to perform our molecular studies as close to "*in vivo"* as possible. To limit the problem of progressive transdifferentiation, we designed all the experiments to end the culture precisely at 9 days after follicle harvesting. The ‘Materials and methods’ section mentions that we passaged the cells at days 5 and 7. After the last passage, cells were either directly seeded for transfection (performed at day 8 and lysis at day 9) or kept in standard culture conditions until day 9 (for ChIP). This was clarified in the ‘Materials and methods’ section.

*2) Another concern is that these investigators did not detect binding of FOXL2 (or ESR1) to the TESCO element upstream of SOX9 which has been mapped in vivo in adult granulosa cells. This suggests that this study may not accurately reflect FOXL2 binding sites in vivo in all cases*.

This was a concern for us too. Several hypotheses can be envisioned. One is that binding of FOXL2 to some of its target sites is indeed lost during cell culture. This is a possibility, as cells progressively change during cell culture and it has previously been shown in other fields that regulatory elements may become more or less accessible during this process. FOXL2 binding to the TESCO element was detected in whole ovaries, so it is also possible that the observed binding occurs in granulosa or theca cells not enriched in our culture system. A possible approach would be to carry ChIP on purified follicles at different stages, and ideally on purified theca and granulosa, which can be technically complex. However, the interesting point here is that we still observe that FOXL2 and estrogens/ESR2 do repress SOX9, even if we do not identify the regulatory element(s) mediating this repression. Thus, our data show that several pathways (and/or elements) are mobilized in this process.

*3) The study derives several key conclusions based upon knockdown experiments. The manuscript, however, does not provide some important experimental details and information on controls. It is evident from the data in*
Figure 1—figure supplement 2
*that specific and selective knockdown has been achieved by each of the siRNAs. The author, however, fail to explain what is used as "control". Ideally, the control cells should express non-specific siRNA and effects of a control siRNA should be validated. In the legend for*
Figure 1—figure supplement 2
*the authors state the following: "levels were normalized by the expression levels of Actb and Sdha". The authors should choose one internal control and specify which was used*.

The control siRNAs used in the study are ON-TARGETPlus “Non targeting pool” from Dharmacon (ref D-001810-10-05), a widely used pool of control siRNAs (see for instance the ref D-001810-10-05 in Google Scholar(scholar.google.com). As for the internal control, the formula we used was unclear. Our internal control is the average relative expression level of Actb and Sdha (corrected in the text). We thought that using several internal controls would add robustness and improve the precision of the baseline. Choosing a control gene is to some extent arbitrary, so we thought picking an average of two generally accepted controls would reduce the level of arbitrariness.